# Adjusting the neuron to astrocyte ratio with cytostatics in hippocampal cell cultures from postnatal rats: A comparison of cytarabino furanoside (AraC) and 5-fluoro-2'-deoxyuridine (FUdR)

**Heiko M. Lesslich**[1]☯*, **Lars Klapal**[1]☯, **Justus Wilke**[1], **Annika Haak**[2], **Irmgard D. Dietzel**[1]

1 Department of Biochemistry II, Ruhr-Universität Bochum, Bochum, Germany, 2 Nanoscopy Group, RUBION, Ruhr-Universität Bochum, Bochum, Germany

☯ These authors contributed equally to this work.
* heiko.lesslich@ruhr-uni-bochum.de

**Data Availability Statement:** All relevant data are within the manuscript and its Supporting information files.

## Abstract

Cell culture studies offer the unique possibility to investigate the influence of pharmacological treatments with quantified dosages applied for defined time durations on survival, morphological maturation, protein expression and function as well as the mutual interaction of various cell types. Cultures obtained from postnatal rat brain contain a substantial number of glial cells that further proliferate with time in culture leading to an overgrowth of neurons with glia, especially astrocytes and microglia. A well-established method to decrease glial proliferation in vitro is to apply low concentrations of cytosine arabinoside (AraC). While AraC primarily effects dividing cells, it has been reported repeatedly that it is also neurotoxic, which is the reason why most protocols limit its application to concentrations of up to 5 μM for a duration of 24 h. Here, we investigated 5-fluoro-2'-deoxyuridine (FUdR) as a possible substitute for AraC. We applied concentrations of both cytostatics ranging from 4 μM to 75 μM and compared cell composition and cell viability in cultures prepared from 0-2- and 3-4-day old rat pups. Using FUdR as proliferation inhibitor, higher ratios of neurons to glia cells were obtained with a maximal neuron to astrocyte ratio of up to 10:1, which could not be obtained using AraC in postnatal cultures. Patch-clamp recordings revealed no difference in the amplitudes of voltage-gated $Na^+$ currents in neurons treated with FUdR compared with untreated control cells suggesting replacement of AraC by FUdR as glia proliferation inhibitor if highly neuron-enriched postnatal cultures are desired.

## Introduction

While experiments performed *in vivo* show the outcome of a particular treatment on the whole organism, cell culture experiments enable researchers to study effects of, for example,

**Funding:** AH was supported by funding from the Deutsche Forschungsgemeinschaft (DFG, German Research Foundation) – project number: 411517989. Furthermore, we acknowledge support by the DFG Open Access Publication Funds of the Ruhr-Universität Bochum.

**Competing interests:** The authors have declared that no competing interests exist.

**Abbreviations:** AraC, Cytosine arabinoside; FUdR, 5-fluoro-2'-deoxyuridine.

the application of defined concentrations of substances for given periods of time. Since brains do not function without glial cells, purified cultures of either cell type offer the unique opportunity to investigate the molecular interactions between glial cells and neurons.

Since its establishment in the 1970s [1], *in vitro* studies of neurons and glial cells have contributed to our understanding of receptor expression and activation of signal cascades in the different cell types of the central nervous system (CNS) in a defined environment. The preparation of neuronal cell cultures is particularly challenging, as neurons do not proliferate, whereas glial cells do. In postnatal primary cell cultures from mammalian brain, the resident glial cells in the CNS (microglia, astrocytes and oligodendrocytes) proliferate with a much higher rate, often overgrowing the neurons [2]. It is therefore necessary to either separate these cell types through, for instance, fluorescent activated cell sorting (FACS), which requires additional immunolabeling and is therefore another source of stress for the cells, or to decrease glial proliferation without decreasing the neuronal survival rate. Furthermore, as both microglia and astrocytes are capable of influencing neurons via secretion of various factors, such as neurotrophic factors (e.g., brain derived neurotrophic factor (BDNF)) [3], growth factors like basic fibroblast growth factor (FGF-2) [4], or cytokines (e.g., tumor necrosis factor-α (TNF-α)) [5], it is favorable to diminish uncontrolled influences of satellite cells. Recently, as an alternative to primary cell culture, the use of induced pluripotent stem cell-derived (iPSC) models has been established. While it is possible to obtain neuron cultures of high purity [6], it is also expensive and time-consuming to generate and characterize an iPSC line. Hence, primary cell culture is still an affordable source for *in vitro* basic research models.

The most commonly used cytostatic to inhibit glial proliferation in primary neuronal cell culture is cytosine arabinoside (AraC) at concentrations of 1 μM to 5 μM (see e.g., [7]). AraC is a cytosine analogue, which is converted to the corresponding triphosphate (araCTP) and subsequently incorporated into the DNA [8]. It inhibits DNA repair [9], thus resulting in DNA fragmentation, and eventually, cell death of mitotic cells. AraC also exhibits a cytotoxic effect on postmitotic neurons [10], which is mediated by oxidative stress through reactive oxygen species (ROS) generation [11]. The use of AraC therefore seems to be restricted to low concentrations, which limits the achievable neuron to glia ratio. A less frequently applied proliferation inhibitor in cell culture investigations is 5-fluoro-2'-deoxyuridine (FUdR, also known as FDUR or floxuridine). While its metabolite 5-fluorouracil (FUra) is incorporated into DNA to some extent and thus might cause DNA replication errors leading to cell death [12], its main mechanism of action differs strongly from that of AraC: FUdR inhibits thymidylate synthase (TS), which causes an imbalance of intracellular deoxyribonucleoside triphosphate (dNTP) pools [13, 14]. This subsequently induces cell death. While as a drug in treatment for neoplastic meningitis, no neurotoxic effect was observed for FUdR [15], in cultures from rat cerebral explants a neuronal death induced by FUdR has been reported [16]. However, in cultured Chinese hamster ovarian cells the potential as antiproliferative agent has been shown to be 20× higher compared to AraC [17], thus making it a promising antimitotic agent for primary neuronal cell culture.

Since we found no systematic investigation of the effects of various dosages of AraC on the cellular composition of postnatal cultures as well as a comparison with FUdR treatments, here we studied the effect of both cytostatics in postnatal primary cell cultures from rats. We varied the concentrations of the two different antimitotic agents from 4 to 75 μM and counted the resulting total cell numbers and βIII-tubulin-positive neurons as well as GFAP-positive astrocytes.

Since the viability of the surviving cells is important for the use of the cultures in further investigations, we additionally studied potential influences of both cytostatics on 3-(4,5-dimethylthiazol-2-yl)-2,5-diphenyltetrazolium bromide conversion to formazan

(MTT-Assay) as an indicator of mitochondrial activity. Finally, we investigated whether the treatment with FUdR, which led to the highest yield of neurons in the cultures, influences sodium currents, which are the main players shaping the upstroke velocity of the action potentials by employing patch-clamp recordings in whole cell configuration.

## Materials and methods

### Ethics statement

Animals were bred in the animal house of the Faculty for Medicine at the Ruhr-Universität Bochum and all procedures adhered to the German animal protection act. Since this publication is based on preliminary work included in the PhD thesis of Lars Klapal some identical, not explicitly cited sentences may be found in this thesis.

### Cell culture

Cell cultures were obtained from brains of postnatal (day 0–4) Wistar-Hannover rats which were sacrificed by decapitation. Rats were divided into two age groups: postnatal day 0–2 (P0-2) and postnatal day 3–4 (P3-4). For each preparation hippocampi of 4–8 animals were collected in ice-cold phosphate buffered saline (PBS, 14190–144, Gibco) that additionally contained 10 mM glucose, 10 mM HEPES, 1 mM pyruvate (11360–070, Gibco), 1 mM glutamine (G7513, Sigma-Aldrich), as well as 25 U/mL penicillin and 25 μg/mL streptomycin (1% P/S; P0781, Sigma-Aldrich). 10 μg/mL desoxyribonuclease I (DN25, Sigma-Aldrich) and 2.5 mg/ mL trypsin (T4549, Sigma-Aldrich) were added. The tissue was incubated under gentle agitation for 7 min at 37˚C. 5% v/v heat inactivated fetal calf serum (FCS, 10270–106, Gibco) was added to stop enzymatic digestion. The suspension was gently triturated 15 times with a 1 mL pipette tip to mechanically dissociate the tissue. After centrifugation of the suspension at 180 g at 4˚C for 10 min, the supernatant was discarded and the pellet was resuspended in 1 mL RPMI 1640 medium (21870–076, Gibco), supplemented with 10% v/v FCS, 1% P/S, and 1 mM glutamine (RPMI⁺). The number of cells in the suspension was evaluated using a Neubauer improved counting chamber. 100,000 cells were seeded into 1 cm diameter glass rings placed in the center of 3.5 cm diameter plastic Petri dishes (Nunc™ 153066, ThermoScientific) which had been coated in advance with 5 μg/mL poly-D-lysine (P6407, Sigma-Aldrich) for 1 h at 37˚C. Cells were incubated at 37˚C and 5% $CO_2$ in a humidified atmosphere in a B5060 incubator (Heraeus).

After one day in vitro, RPMI⁺ medium was exchanged with Neurobasal medium (NB, 21103–049, Gibco) that contained 1% P/S, 1 mM glutamine, and 2% v/v of a custom version of B27 [18] using the recipe of the Hanna research group from the Weizmann Institute of Science in Israel (https://hannalabweb.weizmann.ac.il). Our customized supplement did not include pipecolic acid and we omitted thyroid hormone 3,3',5-Triiodo-L-thyronine (T3) to minimize potential influences on voltage-activated Na⁺ currents by astroglia-secreted factors [19]. Additionally, the medium was supplemented with the cytostatics AraC (C998100, TRC) or FUdR (F0503, Sigma-Aldrich) in various concentrations to be tested. On day 3 in culture, the medium was exchanged with fresh NB medium containing 1% P/S, 1 mM glutamine, and the modified B27 supplement without the addition of any cytostatic. On day 7, experiments were performed.

### Immunocytochemistry

All following steps were carried out at room temperature. Culture media were discarded and cells were fixed in 750 μL paraformaldehyde solution (dissolved to 4% in PBS) for 15 min.

Then, cells were rinsed once with 1 mL PBS. To permeabilize cell membranes and block unspecific binding sites, cell cultures were incubated for 5 min with 750 μL PBS containing 0.1% v/v Triton X-100 and 3% v/v goat serum. This step was repeated once. The supernatant was removed, and cells were rinsed once with 1 mL PBS. Then, cell cultures were incubated for 1 h under gentle shaking with 500 μL of primary antibody solution containing either mouse anti-β3-tubulin (T8660, Sigma-Aldrich) to identify neurons and rabbit anti GFAP (Z0334, DAKO) to label astrocytes. The primary antibodies were applied in dilutions of 1:500 in PBS. Afterwards, the supernatant was discarded, and cultures were rinsed once with 1 mL PBS. Cells were then incubated with 500 μL of secondary antibody solution for 1 h under gentle agitation and protected from light. The secondary antibody solution contained AlexaFluor 488 goat anti-mouse IgG (A11001, Invitrogen) and AlexaFluor 594 goat anti-rabbit IgG (A11012, Invitrogen); both were diluted 1:500 in PBS. Afterwards, the supernatant was removed, and cells were rinsed with 1 mL PBS. To quantify the total number of cells, nuclei were labeled with Hoechst 33258 staining solution (ab228550, Abcam). Cells were incubated for 30 min with 500 μL of 10 μg/mL Hoechst 33258 dye (B2883, Sigma-Aldrich) in PBS under gentle shaking in the dark.

The cultures were examined under an inverted epifluorescence microscope (IX51, Olympus, and IX81, Olympus) with a 20× objective (NA = 0.45; LUCPlanFLN, Olympus). Images were taken with a CCD camera (IX51: XC10, Olympus Soft Imaging Solutions; X81: F-View II, Soft Imaging System) and Olympus cellSens Standard software (version 1.15) at the IX51 microscope and Cell$^M$ software (version 3.1, Olympus) at the IX81 microscope. The total number of cells as well as the numbers of neurons, astrocytes, and unspecified cells of 10 random fields of view with dimensions of 443.8 μm times 334.8 μm were counted and classified using the open-source software CellProfiler (version 4.2.1, Broad Institute, www.cellprofiler.org) [20–23] and CellProfiler Analyst (version 3.0.4, Broad Institute, www.cellprofileranalyst.org) [24–27]. A more detailed description of the applied CellProfiler pipelines and CellProfiler Analyst classification model can be found in the S2 File. Data were obtained from at least three independent preparations.

## MTT assay

Changes in overall mitochondrial activity in the cultures were investigated with the MTT assay, where reduction of 3-(4,5-dimethylthiazol-2-yl)-2,5-diphenyltetrazolium bromide (MTT) to formazan is associated with various metabolic processes in the cells and thus provides a rough indication of the average mitochondrial function [28–30]. For the MTT assay cells were cultured on poly-D-lysine coated 24-well plates at densities of $1 \times 10^5$ cells per well as described in the cell culture paragraph above. On the seventh day *in vitro*, the medium was exchanged with supplemented NB medium additionally containing 0.1 mg/mL MTT (ab146345, Abcam). Cultures were incubated for 3 h at 37˚C and 5% $CO_2$ in a humidified atmosphere. Then, the supernatant was discarded and 1 mL DMSO was added to solve water-insoluble formazan crystals which had formed intracellularly. After 15 min of rigorous shaking, all crystals had been dissolved and absorbance at 570 nm wavelength was measured in a CLARIOstar Plus (BMG Labtech) plate reader. Samples obtained under each condition were evaluated as quadruplicates. Four wells per experiment were measured as blank samples.

## Patch-clamp recordings

Voltage-gated $Na^+$ currents were quantified using whole cell patch-clamp recordings performed at room temperature. The maximum measurement period per cell culture dish was limited to 60 min to ensure comparable measurement conditions for all dishes. Currents were

recorded with a Axopatch 200B amplifier (Molecular Devices) and acquired with the pCLAMP software (version 10.7, Molecular Devices) at sampling rates of 20 kHz. Signals were filtered with a 10 kHz lowpass filter and digitized with an Axon Digidata 1550B analog to digital converter (Molecular Devices). Patch pipettes were pulled from borosilicate glass capillaries (GB-150TF-8P, Science Products) using a PP-830 puller (Narishige). Electrodes had resistances of 2–4 MΩ and were filled with a solution containing (in mM): 0.1 $CaCl_2$, 1.1 EGTA, 5 $MgCl_2$, 5 NaCl, 10 HEPES, and 100 CsF. The extracellular solution contained (in mM): 0.5 $CdCl_2$, 1 $CaCl_2$, 1 $MgCl_2$, 4 4-Aminopyridine, 10 glucose, 10 HEPES, 10 TEA-Cl, and 100 NaCl. Osmolarity of both pipette and extracellular solutions were adjusted to the NB/B22 medium using a semi-micro-osmometer (Knauer). $Na^+$ currents were recorded starting from a holding potential of −77 mV (after liquid junction potential correction) in a series of depolarizing steps in increments of 5 mV. Maximum $Na^+$ current densities were determined as peak currents at a test potential of −17 mV and normalized to membrane capacitance, which was calculated from the integral of the charging curve. Voltage-dependent inactivation of the $Na^+$ currents was determined starting at an initial hyperpolarization of −82 mV and applying a series of prepulse steps of 200 ms duration in 5 mV increments, which were followed by a depolarization step to a test potential of −17 mV to evoke maximal $Na^+$ currents. Peak $Na^+$ currents $I$ were normalized to the current elicited from the most negative prepulse potential $I_0$. $I / I_0$ was plotted as a function of the prepulse potential and fitted to the following Boltzmann equation: $I / I_0 = 1 / (1 + \exp [(V_m − V_{In1/2}) / S])$, where $S$ is the slope, $V_m$ is the prepulse potential, and $V_{In1/2}$ is the prepulse potential at which half of the channels are inactivated. Leakage and capacitive artifacts were subtracted using a P/4 protocol.

Recordings were included in the statistical evaluation if (1) leak currents were smaller than 100 pA; (2) series resistances lower than 20 MΩ; and (3) $Na^+$ currents reached the maximum gradually in dependence on membrane depolarization over a voltage range of at least 20 mV to minimize errors of poor membrane voltage control. A liquid junction potential of 7 mV with respect to the extracellular solution was corrected offline.

### Statistical analysis

Statistical analysis was performed with OriginPro 2021 (OriginLab Corporation). One-way and two-way ANOVA were used to determine statistical significance in data from immunocytochemistry experiments. Tukey's post hoc tests were used to search for significant differences between groups. Box plots show 25% to 75% quartiles. Data points were determined as outliers when they were outside 1.5 times the interquartile range. All data in the texts are represented as means ± SEM.

## Results

### Effects of FUdR and AraC on cell culture composition

Effects of pretreatments with FUdR or AraC were investigated in hippocampal cultures obtained from either postnatal day 0–2 (P0-2) or postnatal day 3–4 (P3-4) rats to include potential effects of the postnatal age at the time of preparation. Cultures were pretreated for 48 h with concentrations of 4, 10, 50 and 75 μM of either AraC or FUdR. Total cell numbers, as identified by staining with Hoechst 33258 dye, were compared for cultures obtained from P0-2 and P3-4 animals after 7 days in vitro (Fig 1). In accordance with previous reports, the treatment with both AraC and FUdR for two days reduced the total cell number significantly (p < 0.001 for all treatments compared to control cultures using ANOVA followed by Tukey's post-hoc test) independent of the concentration of the cytostatic applied. In P0-2 cultures 48-68% less stained nuclei were found, while in P3-4 cultures the total cell number was reduced

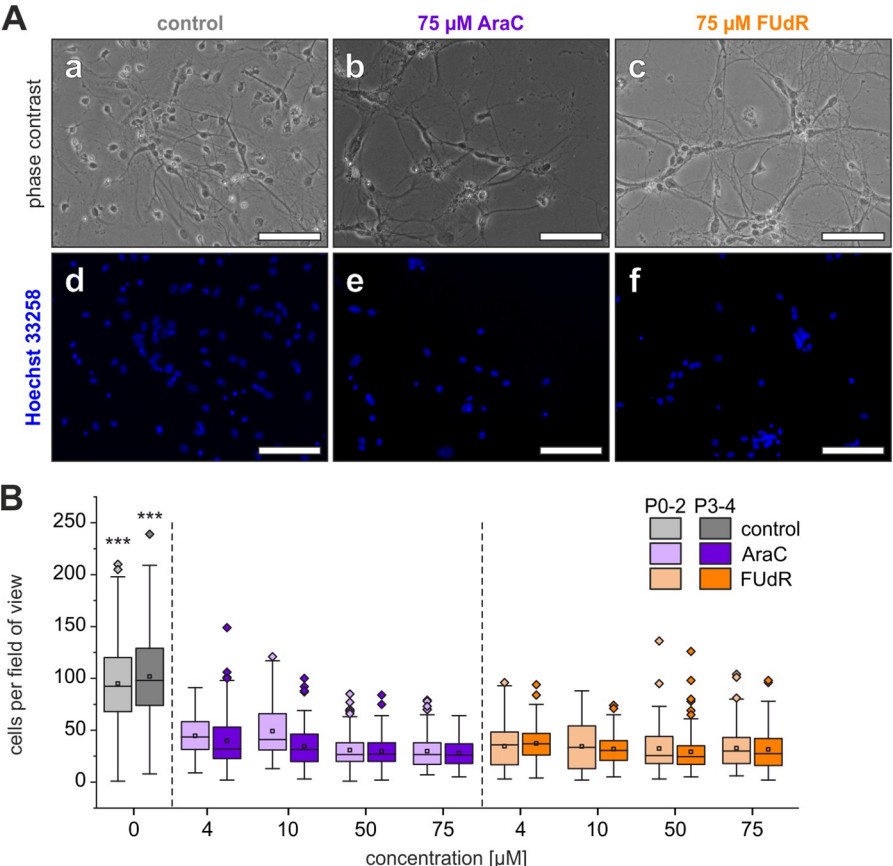

**Fig 1. Concentration- and age-dependent effects of AraC and FUdR on total cell numbers.** (A) Phase contrast micrographs of (a) control cultures and cultures treated with 75 μM of either (b) AraC or (c) FUdR in neural cultures obtained from P0-2 rats. (d-f) Hoechst 33258 staining of the same fields of view. Scale bars represent 100 μm. (B) Number of cells per field of view in cultures obtained from early postnatal (P0-2) and older (P3-4) animals. Boxes represent 25% to 75% quartiles of the data. Squares and lines inside the boxes are means and medians, respectively. Outliers are shown as diamonds. Statistical significances of the decreases in cell numbers with respect to control cultures were $p < 0.001$ (***) for all concentrations.

by 60-71%, compared with untreated cells. However, no significant differences of total cell counts between any of the treated cultures ($p > 0.05$) were found.

In the following, we investigated the effect of different treatments with both cytostatics on the survival of neurons and astrocytes in culture. Compared with control counts of β3-tubulin-positive cells revealed that the number of neurons was not reduced significantly in cultures treated with concentrations up to 50 μM compared with control cultures (Fig 2B). In P0-2 cultures the number of neurons per field of view was significantly reduced from 28.6 ± 1.5 neurons to 16.9 ± 0.9 after treatment with 75 μM AraC ($p < 0.001$). While there was no significant difference between numbers of β3-tubulin positive cells in control and 75 μM FUdR treated cultures ($p > 0.984$), the number of neurons differed significantly after treatment with the two cytostatics at this concentration ($p < 0.001$). In P3-4 cultures we found similar results. The number of neurons was reduced from 26.6 ± 1.2 in control cultures to 20.5 ± 1.1 neurons in cultures treated with 75 μM AraC. This was significant reduction of the proportion of neurons by 23% ($p > 0.038$). Interestingly, in P3-4 cultures there was no significant difference between AraC and FUdR at 75 μM ($p > 0.598$).

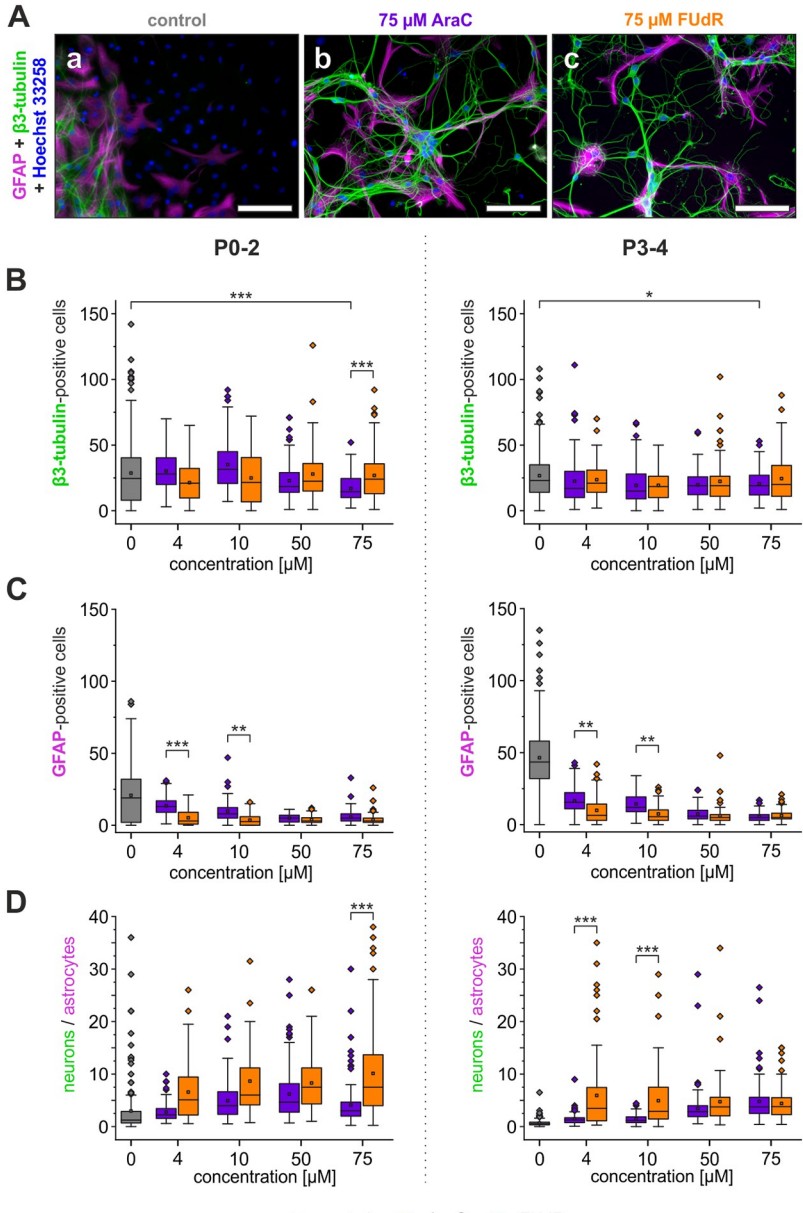

**Fig 2. Quantification of the neuron to astrocyte ratio.** (A) Images taken from (a) control P0-2 cultures and cultures treated with 75 μM (b) AraC or (c) FUdR. Neurons are labeled in green with anti-β3-tubulin antibodies and astrocytes in magenta with anti-GFAP antibodies. Nuclei are visualized in blue by Hoechst 33258 staining. Scale bars represent 100 μm. Concentration-dependent effects of AraC and FUdR on (B) the number of β3-tubulin-positive neurons, and (C) the number of GFAP-positive astrocytes. (D) Ratio of β3-tubulin-positive cells to GFAP-positive cells. Left column shows data obtained from P0-2 cultures, right column data from P3-4 cultures. Data from control cultures labeled in grey, AraC–treated cultures in purple, and FUdR-treated cultures in orange. Boxes represent 25% to 75% quartiles. Squares and lines inside the boxes are means and medians, respectively. Outliers are shown as diamonds. Significant differences in neuron numbers are labeled with * ($p < 0.05$), ** ($p < 0.01$), and *** ($p < 0.005$). Numbers of GFAP-positive astrocytes were consistently significantly smaller with respect to numbers of astrocytes from control cultures ($p < 0.001$) for all cytostatic concentrations tested (not shown in C for clarity).

In contrast to neurons, the number of astrocytes was significantly reduced compared to control cultures following treatment with both cytostatics at all concentrations (p < 0.001). However, at low concentrations of cytostatics (4 and 10 μM), FUdR inhibited astrocyte proliferation more efficiently than AraC resulting in reductions of astrocytes to about 20% of the control level already at a very low concentration of the cytostatic (Fig 2C). The differences between treatments with AraC and FUdR were significant at 4 μM (p < 0.001) and 10 μM (p < 0.0013) in P0-2 cultures and in cultures obtained from P3-4 animals (4 μM: p < 0.006; 10 μM: p < 0.004). A further increase of AraC concentration led to lower astrocytes numbers down to the level obtained after FUdR treatment. At concentrations of 50 μM or higher both antimitotic agents seem to be equally efficient in astrocyte reduction reaching about 20% of control levels.

Fig 2D compares the concentration-dependent effect of AraC and FUdR on the ratio of β3-tubulin-positive neurons to GFAP-positive astrocytes. A significant increase of the neuron to glia ratio was observed (two-way ANOVA, p < 0.001) for all concentrations tested, which was dependent, however, to some extent on the age of the animals at the time of preparation. In P0-2 cultures with increasing concentration of antimitotic agent the neuron to glia ratio increased to a maximum of about 10.1 ± 0.7 neurons per astrocytes after a preincubation with 75 μM FUdR. This was significantly higher than after treatment with 75 μM AraC (p < 0.001). Interestingly, in P3-4 cultures treated with AraC (4–10 μM) the neuron to glia ratio was roughly 1.4 ± 0.1 neuron per astrocyte whereas cultures treated with low concentrations of FUdR showed a significantly higher neuron to astrocyte ratio of about 5.9 ± 0.8neurons per astrocyte (p < 0.001). However, at high concentrations (50 and 75 μM) no significant differences between the effects of the two cytostatics were found in P3-4 cultures.

Whereas in control cultures about 33–38% of all cells were neither GFAP nor β3-tubulin positive, these unidentified cell types almost completely disappeared in cultures treated with an antimitotic agent (Fig 3A). The numbers of unlabeled cells were significantly reduced for all concentrations tested (p < 0.001). In contrast to the differential effect of low concentrations of FUdR and AraC on GFAP-positive astrocytes, both FUdR and AraC were equally effective in inhibiting proliferation of GFAP-negative glial cells with no significant differences between the two cytostatics.

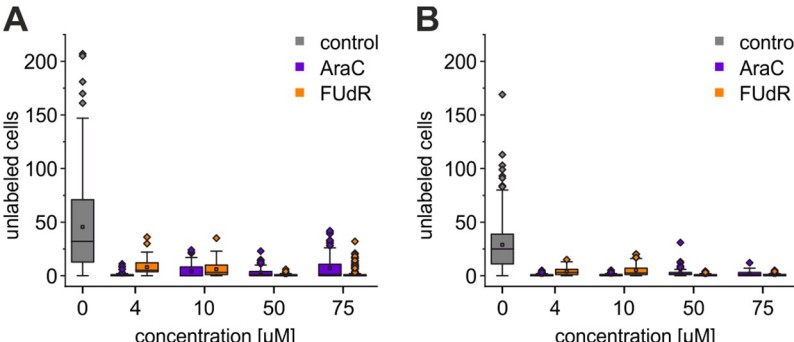

**Fig 3. Effect of AraC and FUdR on the population of unlabeled satellite cells.** Number of unlabeled (GFAP-negative and β3-tubulin-negative) cells in (A) P0-2 and in (B) P3-4 cultures. This population was significantly reduced in all cytostatic-treated cultures (p < 0.001) with respect to control cultures (not indicated by asterisks to enhance clarity). Boxes represent 25% to 75% quartiles of the data. Squares and lines inside the boxes are means and medians, respectively. Outliers are shown as diamonds. Decreases in unlabeled cell counts with respect to control cultures (p < 0.001) were consistently significantly smaller for all cytostatic concentrations tested (not shown for clarity).

## Cell viability at low and high concentrations of cytostatics

The preceding experiments showed that concentrations of 10 μM or lower of FUdR were more efficient than AraC in reducing glia cell numbers. Additionally, in P0-2 cultures treated with 75 μM FUdR significantly more neurons were present compared to 75 μM AraC. Although neurons with elaborate networks were found in cultures pretreated with either AraC or FUdR, we wondered to what extend the metabolic function of the surviving cultures was affected by pretreatment with the cytostatics. We compared MTT assays of P0-2 cultures in untreated cells with those after treatment with 75 μM of AraC and FUdR (Fig 4A). These concentrations were chosen because they had led to the largest differences in glial cell reduction for both cytostatics applied. Optical density at 570 nm was reduced significantly in cultures after pretreatment with 75 μM AraC compared with control cultures (p < 0.006). The absorbance in 75 μM

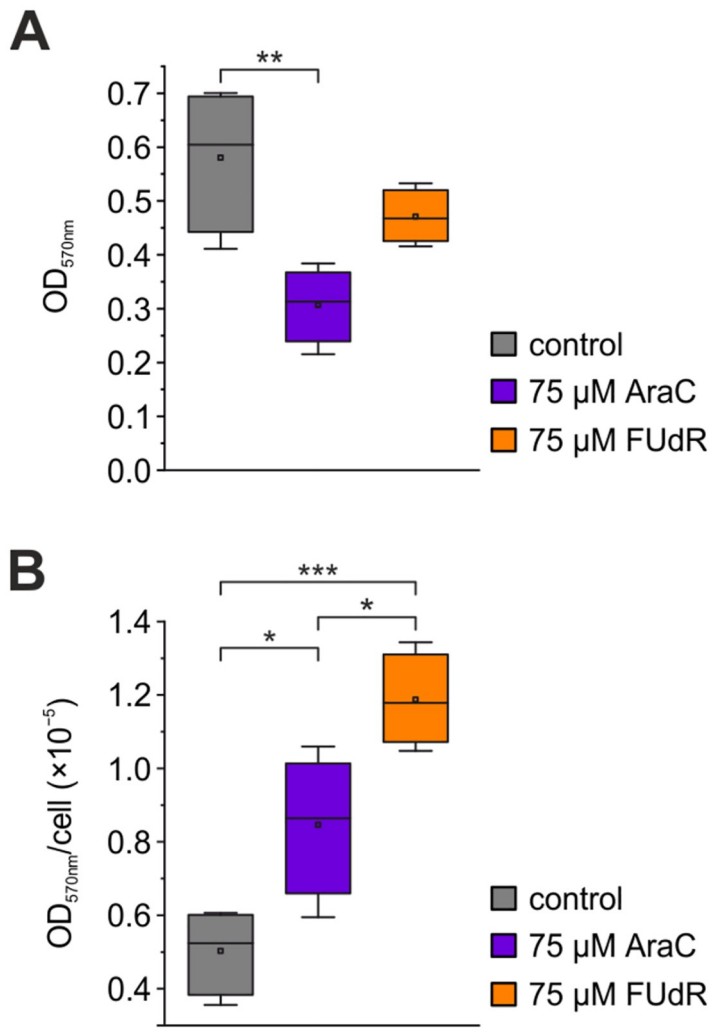

**Fig 4. Assessment of mitochondrial activity after pretreatment with AraC and FUdR using the MTT assay.** (A) Absorbance values at 570 nm, indicating formazan production of untreated cultures and cultures treated with 75 μM of either AraC or FUdR. (B) Absorbance values at 570 nm normalized to the corresponding average total cell numbers taken from Fig 1B. Boxes represent 25% to 75% quartiles of the data. Squares and lines inside the boxes are means and medians, respectively. Outliers are shown as diamonds. Significant differences are marked with * (p < 0.05), ** (p < 0.01), and *** (p < 0.005).

FUdR was 19% lower than in control cultures, however, the effect was not significant ($p > 0.262$). In Fig 4B we normalized the mean absorbance to the corresponding average of total cell numbers using the cell count data from Fig 1B to yield an estimate of the metabolic activity per cell. Interestingly, in cultures treated with either 75 μM AraC or 75 μM FUdR the absorbance per cell was significantly larger compared with the absorbance of control cultures (AraC: $p < 0.023$; FUdR: $p < 0.001$). Moreover, there was a significant difference of the absorbance per cell between AraC and FUdR treated cultures ($p < 0.023$).

## Effects of FUdR on neuronal Na$^+$ currents

While the MTT assay provided an indication of the mitochondrial activity in the cultures, additional patch-clamp investigations were carried out to study potential functional impairments of neurons by a pretreatment with cytostatics. To focus on the most severe effects, we investigated whether a pretreatment with the maximal dosage of AraC and FUdR affected neuronal Na$^+$ currents, which generate the action potential upstroke. Fig 5A shows original recordings of families of sodium currents in pyramid-shaped cells in control and pretreated cultures obtained from P0-2 rats and investigated after a total of 7 days in cultures. In Fig 5B Na$^+$ peak currents normalized to membrane capacitance ($I_{Na}/C$) in bipolar and pyramidal cells are summarized. One-way ANOVA revealed no significant differences in Na$^+$ current densities between treated and untreated cells. We also compared cell membrane capacitances but found no significant differences (Fig 5B). Fig 5D and 5E compare the respective average current versus voltage (I/V)-relationships for bipolar and pyramid-shaped cells in control and pretreated cultures. For both cells types no significant differences between the Na$^+$ current densities of neurons treated with either 75 μM AraC or FUdR were observed in comparison with untreated neurons. Fig 5F and 5G show the voltage-dependence of the inactivation of the Na$^+$ channels. No significant differences between the steady-state inactivation of Na$^+$ channels of the treated and untreated bipolar and pyramid-shaped neurons were observed.

## Discussion

### Treatment with AraC and FUdR increases the neuron to astrocyte ratio

To investigate neuron-glia interactions as well as to study pharmacological effects on different cell types in isolation, cultures with defined neuron to glia ratio are desirable. Treatment of neural cultures with the antimitotic agents AraC and FUdR are known to efficiently inhibit glial proliferation in vitro as has been shown in previous studies [16, 31]. In protocols for preparation of primary neuronal cultures AraC is added in most cases as a mitotic inhibitor to limit glial proliferation. However, not only astrocytes and other types of glial cells are affected by AraC treatment but neuronal cell death elicited by oxidative stress has been reported to be caused by cytostatics as well [11]. Our present findings show that AraC can be used at much higher concentrations than suggested in many cell culture protocols. AraC concentrations up to 50 μM for 48 h did not lead to significantly decreased numbers of neurons after 7 days in culture under the present conditions. This is in good agreement with results from one report in which the EC$_{50}$ of AraC has been quantified to be 60 μM [32]. Differences from our results to most other studies that report high neurotoxicity might be explained by the formulation of the culture medium. AraC neurotoxicity has been shown to be mediated by ROS and this effect was demonstrated to be dependent on glutathione [11]. In the present study, serum containing RPMI was exchanged to a defined neurobasal medium after one day in vitro. This medium contained among others the antioxidants superoxide dismutase and glutathione, which might have been able to eliminate ROS generated upon AraC treatment. However, Dessi et al. determined the EC$_{50}$ of AraC to be 60 μM in cerebellar neurons obtained from 8-day-old rats

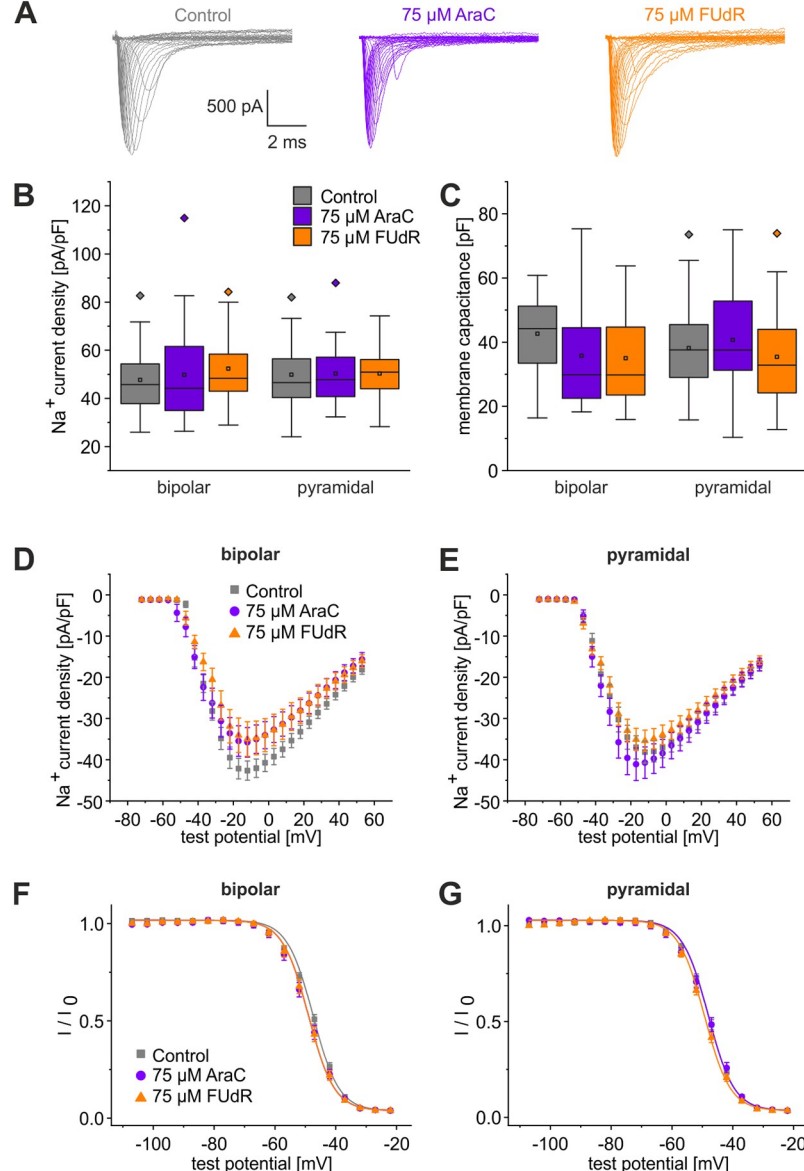

**Fig 5. Comparison of Na⁺ current recordings in control cultures compared with cultures treated with FUdR or AraC prepared from P0-2 rat pups.** (A) Series of original Na⁺ current recordings from pyramidal neurons elicited by step depolarizations in 5 mV increments starting from a holding potential of −77 mV. (B) Peak Na⁺ current densities of neurons from control cultures and from neurons cultured in the presence of either 75 μM AraC or 75 μM FUdR for two days measured at test potentials of −17 mV starting from holding potentials of −77 mV. (C) Cell membrane capacitances calculated from the area beneath the capacitive current after applying a test potential of 20 mV from a holding potential of −77 mV. Average current versus voltage relationships for Na⁺ currents normalized to capacitance. of (D) bipolar and (E) pyramidal cells. Steady-state inactivation of Na⁺ currents in control, AraC and FUdR treated cultures recorded from (F) bipolar and (G) pyramidal cells. Solid lines represent fits to the Boltzmann equation (see Materials and methods). Recordings from control cells: grey squares, 75 μM AraC: purple circles, 75 μM FUdR: orange triangles. Boxes represent 25% to 75% quartiles of the data. Squares and lines inside the boxes are means and medians, respectively. Outliers are shown as diamonds. No significant differences were found. Data obtained from at least 15 cells per condition from 5 independent preparations.

cultured in serum-containing medium without further addition of antioxidants [32]. For FUdR we did not observe any significant reduction in neuronal cell. Even at 75 μM FUdR no changes in numbers of neurons were detected. This finding contradicts previous observations [16], where a decrease in the number of neurons up to ~30% of control was observed in cerebral explants from 20-day old fetal rats after application of 10 μM FUdR on days 4-7 in vitro with measurements performed after 18 days in vitro. It remains to be investigated, whether the increased application period of 10 μM FUdR, the longer culture period or the age of the rats, at which the preparation took place, are the cause for the differences of the results.

Although it has been reported that the ratio of neuronal to non-neuronal cells in the hippocampus of rats at postnatal days 0 to 4 only scarcely changes [33], we subdivided rat preparations into two age groups to investigate an influence of the preparation day on culture composition. Our findings show that even a delay of two days of the time point of preparation might lead to slightly different cell culture compositions. In P3-4 cultures, the addition of 75 μM FUdR in various concentrations led to results similar to that of P0-2 cultures, that is numbers of neurons were reduced by about 10%. In contrast, in cultures pretreated with 75 μM AraC neuron numbers compared to control were up to 40% smaller in P0-2 cultures while in P3-4 cultures a reduction by maximally 28% was observed. Not only might the age of the prepared animals exert a major influence on the neurotoxicity of AraC, but also the time of mitotic inhibitor addition, antioxidants in the culture medium and the incubation period. Although we cannot fully explain the discrepancies of the studies published so far, our results show, that AraC is tolerated by neurons up to concentrations of 50 μM in our cultures and that below this concentration no significant neurotoxic effect was found. The most significant finding of our present study is, that FUdR reduces the number of GFAP-positive astrocytes consistently at much lower concentrations than AraC, leading to neuron to astrocyte ratios of up to 10:1. In cultures treated with low concentrations of FUdR ($< 25$ μM) the number of neurons per astrocytes are at least twice as high compared to AraC treatment.

We observed that in control cultures nearly 33–38% of all cells were neither β3-tubulin- nor GFAP-positive. This percentage decreased in cultures after treatment with cytostatics to less than 5% (Fig 3). To what extend GFAP-negative astrocytes [34], activated microglia, or oligodendrocytes and their precursors contribute to the unlabeled cell population had not been clarified in the present study. However, it is remarkable that even low concentrations of cytostatics nearly extinct these cell types. Furthermore, it should be kept in mind, that the low osmolarity of the neurobasal culture medium used in the present study contributed to decrease the number of oligodendrocytes in the present cultures [35]. Irrespective of the exact composition of the β3-tubulin- and GFAP-negative cell population we observed no significant differences between the numbers of the 5% surviving residual cells following treatment with either cytostatic.

## Effects of the pretreatment with the cytostatics on average metabolic activity and sodium current density in the cultures

We further investigated whether AraC and FUdR treatments exerted conspicuous effects on mitochondrial function in cultures, which could signal impairments of cellular viability. Cultures treated with cytostatics showed lower numbers of cells compared with control cultures (Fig 1). This explains the lower formazan production observed in cultures treated with cytostatics compared to control cultures. Whereas no significant difference in the reduction of total cell numbers were observed between AraC and FUdR at 75 μM, a decrease in optical density of the reaction product was only significant in cultures treated with 75 μM AraC ($p < 0.006$). Metabolic activity measured as change in optical density by conversion of MTT to

formazan correlates to the number of cells [36] and is therefore sometimes used to estimate the cell number in cultures. A comparison with the results shown in Fig 1 reveals, that the more time-consuming direct cell counts offer much more sensitive and significant results, however. If the MTT assay only reflects the number of cells in the culture under the presently studied conditions, a normalization to cell counts under the different culture conditions should result in equal numbers. Interestingly, the treatment with cytostatics resulted higher formazan production per cell, which was especially prominent after treatment with 75 μM FUdR (p < 0.023). This effect might be explained by the higher percentage of excitable neurons in the culture which need to produce more ATP to decrease their $Na^+$-load.

Since our observations showed, that a pretreatment with 75 μM FUdR led to significantly more neurons and tendentially preserved metabolic function better than a treatment with AraC, we performed further investigations on potential effects of the cytostatics on functional properties of the neurons, recorded after 7 days in culture. Among many potential parameters, which could be investigated we concentrated on the comparison of voltage-activated sodium currents from AraC- or FUdR-treated to control neurons, since these currents are the main determinants of action potential upstroke velocity. To further differentiate between the main neuronal subtypes, we split the recorded cells into two groups: bipolar, as characteristic for inhibitory, and pyramid-shaped neurons, representing potentially excitatory neurons. No statistically significant differences concerning sodium current density, cell membrane capacitance, which is a measure of the soma-near membrane surface, and current versus voltage relationships as well as voltage-dependence of inactivation were observed in both bipolar and pyramidal cells, indicating that treatment with either AraC or FUdR for 48 h did not change the functional properties and soma size of the cells.

## Conclusion

Primary neuronal cell culture is an intensively used tool to investigate the effects of drugs on purified cell populations. We compared the efficiency of two antimitotic agents to minimize glia content in a dose-dependent manner to obtain highly enriched neuronal cultures. To the best of our knowledge, we present the first direct comparison of the effects of AraC and FUdR on the neuron to astrocyte ratio in neuronal cell cultures obtained from hippocampi from postnatal rats. In cerebellar neurons from postnatal rats the $EC_{50}$ of AraC has been found to be 60 μM [32]. However, in commonly used protocols only much lower concentrations of AraC are used (see e.g., [37–39]). Under our presently used culture conditions containing antioxidants, AraC was used to enrich neurons without exerting significant neurotoxic effects up to concentrations of 50 μM, resulting in up to 6 neurons per astrocytes. In contrast, at low concentrations FUdR was much more effective at inhibiting glial proliferation leading to about 50% less astrocytes than in AraC treated cultures, thus resulting in even purer postnatal neuron cultures with neuron to astrocyte ratios of up to 10:1. Both cytostatics equally inhibited the non-GFAP-positive glia populationfrom 33–38% to less than 5% of total cell count.

In conclusion, FUdR seems to be the cytostatic of choice when high neuron to astrocyte ratios are required. Additionally, no significant reduction of neurons was observed in FUdR-treated cultures compared to control cultures even at the highest concentration investigated in the present study. In case that experimental conditions require mixed cultures, low concentrations of AraC are preferred yielding neuron to astrocyte ratios of 1:1 to 2:1. Thus, the desired composition can be adjusted by varying antimitotic agents and their concentration. Despite previous studies reporting high neurotoxicity elicited from AraC, we were able to increase the neuron to astrocyte ratio with AraC to a maximum of 6 neurons per astrocyte at 50 μM

without significant neuronal loss. The reasons for this contradicting results, however, have to be explored in futures studies.

## Supporting information

**S1 File. Minimum dataset.**
(XLSX)

**S2 File. Description of CellProfiler pipelines and CellProfiler Analyst classification model.**
(ZIP)

## Acknowledgments

We thank Stephanie Koll and Heidrun Breuker-Siraj for assistance in the laboratory, the RUBION for granting us access to their IX81 fluorescence microscope and Thomas Günther-Pomorski for the support of our research group.

## Author Contributions

**Conceptualization:** Heiko M. Lesslich, Lars Klapal, Annika Haak.

**Data curation:** Heiko M. Lesslich.

**Formal analysis:** Heiko M. Lesslich, Lars Klapal, Justus Wilke, Annika Haak.

**Investigation:** Heiko M. Lesslich, Lars Klapal, Justus Wilke.

**Project administration:** Heiko M. Lesslich, Lars Klapal, Irmgard D. Dietzel.

**Supervision:** Irmgard D. Dietzel.

**Visualization:** Heiko M. Lesslich, Lars Klapal.

**Writing – original draft:** Heiko M. Lesslich, Lars Klapal.

**Writing – review & editing:** Heiko M. Lesslich, Lars Klapal, Justus Wilke, Annika Haak, Irmgard D. Dietzel.

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
