## [Decision Letter · Decision Letter 0]

21 Oct 2021

PONE-D-21-30899Adjusting the neuron to astrocyte ratio with cytostatics in hippocampal cell cultures from postnatal rats: A comparison of cytarabino furanoside (AraC) and 5-fluoro-2’-deoxyuridine (FUdR)PLOS ONE

Dear Dr. Leßlich,

Thank you for submitting your manuscript to PLOS ONE. Your work has been evaluated by two experts in the field.  After careful consideration, we feel that it has merit but does not fully meet PLOS ONE’s publication criteria as it currently stands. Therefore, we invite you to submit a revised version of the manuscript that addresses the points raised during the review process.

In your revised manuscript, please be sure to address technical and substantive comments of both reviewers on:Potential bias of cell count procedures (via either detailed information on blinding procedures or validation with automated counts)Quantitative/statistical validity of your conclusions (several comments by both reviewers, which are included below)Your ability to discriminate between two markers whose fluorescence appears to be captured in the same channel

We look forward to receiving your revised manuscript.

Kind regards,

Alexander A. Mongin, Ph.D.

Academic Editor

PLOS ONE

Journal Requirements:

2. If anesthesia, euthanasia, or any kind of animal sacrifice is part of the study, please include briefly in your statement which substances and/or methods were applied.

"AH was supported by funding from the Deutsche Forschungsgemeinschaft (DFG, German Research Foundation) ‒ project number: 411517989. Furthermore, we acknowledge financial support by the Open Access Publication Funds of the

Ruhr-Universität Bochum."

Reviewers' comments:

Reviewer's Responses to Questions

**Comments to the Author**

1. Is the manuscript technically sound, and do the data support the conclusions?

Reviewer #1: Partly

Reviewer #2: Yes

2. Has the statistical analysis been performed appropriately and rigorously? 

Reviewer #1: No

Reviewer #2: Yes

3. Have the authors made all data underlying the findings in their manuscript fully available?

Reviewer #1: Yes

Reviewer #2: Yes

4. Is the manuscript presented in an intelligible fashion and written in standard English?

Reviewer #1: No

Reviewer #2: Yes

5. Review Comments to the Author

Reviewer #1: The authors of this paper investigated the use of AraC and FUdR at varying concentrations on reducing numbers of glia in cultures from P0-2 and P3-4 rats. They claim that FUdR is favorable at reducing glial cell numbers at low concentrations while increasing the number of neurons as compared to AraC. They also demonstrate that using FUdR at high concentrations doesn’t impact neuronal cell function through Na+ current measurements.

This study focuses on an interesting and relevant area that is useful for in vitro studies looking at neuron-glia interactions. The study aims to address an issue in the field, where reductions in glia in cell cultures are necessary to prevent their overgrowth and to prevent reductions in neuronal cells. The methods used in the study are scientifically sound, however, the claims made from the findings are not supported by the data and statistical analysis. The background information relating to what is known about FUdR, specifically the caveats, is lacking. The discussion also fails to expand upon some of their assumptions and how other literature supports these conclusions. Despite my disagreement with their major findings, I believe the addition of key experiments may help to support some of these claims. However, the likely major finding of this manuscript is on reporting that AraC is not cytotoxic at higher concentrations when using antioxidants but this would need to be further developed.

Major points:

Fig. 2:

1. Authors mention that astrocyte and neuron morphology are not affected and then show representative images but give no further information on how that was determined. It would be helpful to describe how this was decided (i.e. specific morphological features, size, etc.). Quantifying morphology would increase the rigor for this claim.

2. The authors frequently mention that neuronal numbers are increased in Fig 2B, however, no significant differences were shown. There may be a trend for increase in the P3-4 age cultures at low concentrations for FUdR, but this distinction is not made throughout the text.

3. The authors claim that 4 μM FUdR reduces GFAP+ cells more significantly than AraC, however, they fail to distinguish that this is only the case in the P3-4 cultures but not for the P0-2 cultures.

4. Overall, FUdR and AraC treatment appear to be equally effective at all concentrations for P0-2 except when comparing the neuron to astrocyte ratio (this difference isn’t seen on individual numbers of neurons in 2B and astrocytes in 2C). The only notable difference in cell numbers appears to be the effect on reducing astrocytes at low concentrations of AraC at P3-4. What could explain this discrepancy in no change in neurons and astrocytes in 2B and 2C, yet a significantly higher ratio of neurons vs. astrocytes in 2D?

5. The text states “In P0-2 cultures with increasing concentration of antimitotic agent the neuron to glia ratio increased to a maximum of about 6:1” suggesting that this is a dose-dependent mechanism but the data does not support this claim (significant differences at 10 μM and again at 75 μM).

6. Perhaps the most important finding of the paper lies within these graphs, and it’s that AraC is not effective at reducing astrocytes at low concentrations only in older P3-4 cultures, but at high concentrations at all ages, it is effective and doesn’t appear to be cytotoxic to neurons. This information is valuable and needs to be further explored.

Fig. 3:

1. Authors state that “unlabelled cells almost disappear after treatment” which is not followed up by any quantification or p-value.

2. Quantification of microglia was done using TMEM119, however, in the figure it is stated that both BIII-tubulin and TMEM119 are both in green. How were they separated for analysis if under the same channel?

3. The text discusses reductions in microglia and GFAP negative glial cells but doesn’t have any graph or data supporting this. Please include this figure.

4. The text states “The preceding experiments showed that concentrations of 10 μM or lower of FUdR were more efficient than AraC in reducing glia cell numbers” this is only true in the case of P3-4 cultures, but not for P0-2 cultures.

5. The discussion states “Judged by morphology, the further unlabeled cells in our cultures might be GFAP-negative astrocytes [28], activated microglia, less well stained with TMEM119, or oligodendrocytes and their precursors.” This can be determined using other markers for these cell types, including NG2 for OPCs, A2B5 for pre-oligos, and Iba1 or CXCL4 for microglia. Also, TMEM119 is developmentally regulated and doesn’t arise until later postnatal time points up to P14 so some negative cells can exist, whereas Iba1 or CXCL4 will identify all microglia (Bennett, ML et al., 2016, PNAS).

Fig. 4:

1. The text states “There was, however, an insignificant tendency for the highest absorbance per cell in the cultures treated with 50 μM FUdR, which contain the highest number of neurons (see Fig 2D).” this isn’t supported by the data in 2B which quantifies numbers of neurons.

2. The authors report that 50 μM FUdR treatment condition causes increased formazan production which does not reach statistical significance. They claim that this may be due to increased mitochondrial activity in neurons due to Na+ load from higher neuronal cell numbers. As stated previously, the data does not support that there is an increase in neuronal cells, however there is indeed a reduction in glia. Including an additional analysis of the higher FUdR concentrations will be helpful to see if any changes to overall cell viability are occurring. Furthermore, investigating the effects on astrocytes as a whole is essential, including the possibility of excitotoxicity due to reduced astrocyte coverage.

Fig. 5:

1. The authors only investigated the effects of FUdR on the Na+ currents in these cultures but did not look at the effect of AraC. The text is lacking in rationale for why they left this out. It is especially relevant given that the field believes AraC to be cytotoxic at higher doses, where this paper proves this not to be the case.

Minor points:

6. Fig. 1: The text says that you look at 6 μM for AraC and 25 μM for FUdR but then figure 1 shows all concentrations and throughout the rest of the text, 6 μM for AraC is never mentioned again.

7. Fig 2 : The text says that AraC is in orange and FUdR is in purple but in 2A, the text shows AraC in purple and FUdR in orange.

8. Fig 3: Fig 3E, pink arrow on bottom right is pointing to a GFAP positive cell.

9. Fig 4: Figure legend for 4A says “shown as circles” but there are no circles.

Reviewer #2: In this manuscript the authors proposed FUdR as an alternative cytostatic that can be used instead of AraC to increase the ratio of neurons to astrocytes. They compared different doses of both treatments and found that at high doses both cytostatic have similar effects, but at low doses FUdR is more effective in killing astrocytes and therefore enriching neuronal cultures.

The research is important in the field as in vitro tools are still required to evaluate molecular mechanisms. There are, however, some weakness in the analysis of the data and some changes are suggested to improve quality of the manuscript.

1. In financial disclosure: Is stated that the author(s) received no specific funding for this work, however in acknowledge section some funding is mentioned

2. The cells were manually counted, and the cell numbers is key to the conclusions of the paper, the reviewer would suggest either an automatic cell counting (several plugins published for Image J allow automatic cell counting), a clear description of how the analyzer was blind to the culture condition, or ideally both

3. Add was found at the end of first paragraph of results… however, no significant differences of total cell count between any of the treated cultures was found (p>0.05).

4. Figure legend 1. Should indicate how many independent cultures were analyzed instead of numbers of dishes

5. Pooling data together from different drug concentrations is not a valid comparison and should be removed

6. I recommend changing the color of the ICC-IF since red, green colour blindness is the most common form of colour vision deficiency

7. Page 15, line 274 Treatment with 4 uM FUdR, however, inhibited astrocyte proliferation more efficiently than AraC in cultures prepared from P3-4 rats, resulting in a reduction of astrocytes to about 20% of the control level.

It needs to specify P3-4 since the difference is not significant for P0-2 culture, also eliminate already at a very low concentration of the cytostatic as the concentration is already mentioned.

8. Page 15, lines 284-286. There was “almost” no difference between the two age groups… or there was a significant age-dependent effect??

9. Page 16, line 287. In P0-2 cultures with increasing concentration of antimitotic agent the neuron to glia… be specific, replace antimitotic agent by FUdR

10. Using the same color for two different markers (in this case, TMEM119 and beta3-tubulin) is not acceptable. The authors need to repeat this staining if the same secondary was used or image the ICC-IFs again using a microscope with more cubes or lasers to be able to separate the 4 markers used.

11. Re-write page 17 lines 318-320 to improve readability. The sentence “To assess cell…” is repetitive and unclear

12. Remove sentence “There was, however, an insignificant tendency…” in page 17 lines 328. Insignificant tendency = no difference

13. Figure legends 4 and 5 states “shown as circles/single data point plotted as circles” no circles are shown in the figures

14. Page 23 lines 452-455, and 456-457. The authors make conclusions based on no significative effects. Slightly higher (p>0.1856), slightly improved, tendentially preserved… These need to be removed from the text as are not statistical significative effects and only distract the reader of the main conclusion of the paper that is clearly stated in Page 21 lines 416-418.

15. In conclusion, page 25, line 473 replace “minimize interaction with glia” for “minimize glia content” and “highly purified neuronal cultures” for “highly enriched neuronal cultures”

16. Page 25, line 489 replace “experimental conditions require more astrocytes to be present in the neuronal vicinity, low…” for “experimental conditions require mix cultures, low…”

6. PLOS authors have the option to publish the peer review history of their article (what does this mean?). If published, this will include your full peer review and any attached files.

Reviewer #1: No

Reviewer #2: No

---

## [Author Response · Author response to Decision Letter 0]

27 Jan 2022

Rebuttal letter

Dear editor, dear reviewers,

first of all, and most importantly, we want to thank you for thoroughly reviewing our manuscript and giving us constructive feedback. The reviewers’ comments were more than helpful and we feel they greatly improved our manuscript: After revising the text, refining the data and statistical analysis, and by repeating some of the experiments, we were able to improve the quality of our manuscript and hope that it is now suitable for publication in your journal. 

In the following, we address the editor’s and reviewers’ comments. We hope that we explained all changes and answered all questions sufficiently. Thank you again for the constructive feedback.

Kind regards

Heiko Leßlich

 

Editor: Please ensure that your manuscript meets PLOS ONE's style requirements, including those for file naming.

Authors: We checked all file names and we have followed the style requirements given by PLOS ONE to the best of our abilities.

Editor: If anesthesia, euthanasia, or any kind of animal sacrifice is part of the study, please include briefly in your statement which substances and/or methods were applied.

Authors: We have included a statement mentioning that rats were killed by decapitation.

Editor: We note that you have provided funding information that is not currently declared in your Funding Statement. However, funding information should not appear in the Acknowledgments section or other areas of your manuscript. We will only publish funding information present in the Funding Statement section of the online submission form. 

Authors: We have removed funding information from the manuscript and included a request in the cover letter to change the funding statement.

Editor: Your ethics statement should only appear in the Methods section of your manuscript. If your ethics statement is written in any section besides the Methods, please move it to the Methods section and delete it from any other section. Please ensure that your ethics statement is included in your manuscript, as the ethics statement entered into the online submission form will not be published alongside your manuscript. 

Authors: We have moved the ethics statement to the Methods section.

Editor: Please review your reference list to ensure that it is complete and correct. If you have cited papers that have been retracted, please include the rationale for doing so in the manuscript text, or remove these references and replace them with relevant current references. Any changes to the reference list should be mentioned in the rebuttal letter that accompanies your revised manuscript. If you need to cite a retracted article, indicate the article’s retracted status in the References list and also include a citation and full reference for the retraction notice.

Authors: We added some more information about FUdR in the introduction. Therefore, one additional reference was cited:

 Senderoff RI, Weber PA, Smith DR, Sokoloski TD (1990) Evaluation of antiproliferative agents using a cell-culture model. Investigative ophthalmology & visual science 31 (12): 2572–2578.

Authors: We cited several reports in the methods section about CellProfiler and CellProfiler Analyst. As these tools were used to automatically analyze the immunocytochemical stained images:

 Stirling DR, Swain-Bowden MJ, Lucas AM, Carpenter AE, Cimini BA et al. (2021) CellProfiler 4: improvements in speed, utility and usability. BMC bioinformatics 22 (1): 433.

 McQuin C, Goodman A, Chernyshev V, Kamentsky L, Cimini BA et al. (2018) CellProfiler 3.0: Next-generation image processing for biology. PLoS biology 16 (7): e2005970.

 Kamentsky L, Jones TR, Fraser A, Bray M-A, Logan DJ et al. (2011) Improved structure, function and compatibility for CellProfiler: modular high-throughput image analysis software. Bioinformatics (Oxford, England) 27 (8): 1179–1180.

 Carpenter AE, Jones TR, Lamprecht MR, Clarke C, Kang IH et al. (2006) CellProfiler: image analysis software for identifying and quantifying cell phenotypes. Genome biology 7 (10): R100.

 Stirling DR, Carpenter AE, Cimini BA (2021) CellProfiler Analyst 3.0: Accessible data exploration and machine learning for image analysis. Bioinformatics (Oxford, England).

 Jones TR, Carpenter AE, Lamprecht MR, Moffat J, Silver SJ et al. (2009) Scoring diverse cellular morphologies in image-based screens with iterative feedback and machine learning. Proceedings of the National Academy of Sciences of the United States of America 106 (6): 1826–1831.

 Dao D, Fraser AN, Hung J, Ljosa V, Singh S et al. (2016) CellProfiler Analyst: interactive data exploration, analysis and classification of large biological image sets. Bioinformatics (Oxford, England) 32 (20): 3210–3212.

 Jones TR, Kang IH, Wheeler DB, Lindquist RA, Papallo A et al. (2008) CellProfiler Analyst: data exploration and analysis software for complex image-based screens. BMC bioinformatics 9: 482.

Authors: The following reference has been removed because the experiments using the fixation method described by Richter et al and labeling with TMEM199 are no longer part of the manuscript:

 Richter KN, Revelo NH, Seitz KJ, Helm MS, Sarkar D et al. (2018) Glyoxal as an alternative fixative to formaldehyde in immunostaining and super-resolution microscopy. The EMBO journal 37 (1): 139–159.

 Bennett ML, Bennett FC, Liddelow SA, Ajami B, Zamanian JL et al. (2016) New tools for studying microglia in the mouse and human CNS. Proceedings of the National Academy of Sciences of the United States of America 113 (12): E1738-46.

Reviewer #1: Fig. 2: Authors mention that astrocyte and neuron morphology are not affected and then show representative images but give no further information on how that was determined. It would be helpful to describe how this was decided (i.e. specific morphological features, size, etc.). Quantifying morphology would increase the rigor for this claim.

Authors: We did not perform any quantitative analysis of morphological features. We do not think that cell size, number of processes, or their length have changed significantly in our cultures. However, we understand that we shouldn’t make statements about cell morphology if there was no quantitative morphology analysis. We omitted all statements regarding cell morphology. A quantitative analysis of such features might be interesting in a future project in combination with scanning probe microscopy techniques.

Reviewer #1: Fig. 2: The authors frequently mention that neuronal numbers are increased in Fig 2B, however, no significant differences were shown. There may be a trend for increase in the P3-4 age cultures at low concentrations for FUdR, but this distinction is not made throughout the text.

Authors: We removed all statements mentioning that neuronal numbers are increased as they were not statistically significant.

Reviewer #1: Fig. 2: The authors claim that 4 μM FUdR reduces GFAP+ cells more significantly than AraC, however, they fail to distinguish that this is only the case in the P3-4 cultures but not for the P0-2 cultures.

Authors: After re-analyzing the data with CellProfiler (Analyst), we found that 4 µM FUdR is more efficient than 4 µM AraC in reducing GFPA+ cells in both P0-2 and P3-4 cultures. 

Reviewer #1: Fig. 2: Overall, FUdR and AraC treatment appear to be equally effective at all concentrations for P0-2 except when comparing the neuron to astrocyte ratio (this difference isn’t seen on individual numbers of neurons in 2B and astrocytes in 2C). The only notable difference in cell numbers appears to be the effect on reducing astrocytes at low concentrations of AraC at P3-4. What could explain this discrepancy in no change in neurons and astrocytes in 2B and 2C, yet a significantly higher ratio of neurons vs. astrocytes in 2D?

Authors: This discrepancy is due to the nature of fractions. Even though the differences between AraC and FUdR treated cultures concerning the numbers of GFAP+ and betaIII-tubulin+ cells were marginal (hundreds), this small margin has a big impact on the calculated fractions/ratios. From 50 to 75 µM neuron counts changed in AraC (-900) and FUdR (+2000) treated cultures, while simultaneously astrocyte counts decreased by 300 in each culture treatment condition. Therefore, the neuron per astrocyte ratio effectively increased in the FUdR treated cultures. However, after re-evaluating the data with CellProfiler (and CellProfiler Analyst), we also found a statistically significant reduction in the neuron numbers for 75 µM AraC.

Reviewer #1: Fig. 2: The text states “In P0-2 cultures with increasing concentration of antimitotic agent the neuron to glia ratio increased to a maximum of about 6:1” suggesting that this is a dose-dependent mechanism but the data does not support this claim (significant differences at 10 μM and again at 75 μM).

Authors: We agree with the reviewer and have removed this statement.

Reviewer #1: Fig. 2: Perhaps the most important finding of the paper lies within these graphs, and it’s that AraC is not effective at reducing astrocytes at low concentrations only in older P3-4 cultures, but at high concentrations at all ages, it is effective and doesn’t appear to be cytotoxic to neurons. This information is valuable and needs to be further explored.

Authors: After re-analyzing the data with CellProfiler and CellProfiler Analyst the results slightly changed. The results from P3-4 cultures are now consistent with those obtained from P0-2, i.e. AraC seems to be indeed slightly neurotoxic at the highest concentration (75 µM) investigated in the present study. However, AraC is significantly less efficient in reducing GFAP+ cells than FUdR at 4 µM. Yet, both cytostatics significantly reduced GFAP+ cells compared with control.

Reviewer #1: Fig. 3: 1. Authors state that “unlabelled cells almost disappear after treatment” which is not followed up by any quantification or p-value.

Authors: We added a p-value.

Reviewer #1: Fig. 3: 2. Quantification of microglia was done using TMEM119, however, in the figure it is stated that both BIII-tubulin and TMEM119 are both in green. How were they separated for analysis if under the same channel?

Authors: TMEM119-labeled cells were not quantified. The experiments in Fig3B were just meant to indicate that almost all of the previously unlabeled cells were likely microglia. If nearly all GFAP- and beta-III-tubulin-negative (unlabeled) cells had disappeared after an additional TMEM119 staining, it would have indicated that most of the unlabeled cells in Fig2A must have been microglia. However, we understand that this experiment is too preliminary and confusing and have removed it from the manuscript.

Reviewer #1: Fig. 3: To 1 and 2: 3. The text discusses reductions in microglia and GFAP negative glial cells but doesn’t have any graph or data supporting this. Please include this figure.

Authors: We have not gone into a detailed analysis of the residual cell types and reduced Figure 3 A (p0-2) and B(p3-4) to show the changes in unlabeled, non GFAP, non bIII-tubulin positive cells after different treatment conditions.

Reviewer #1: Fig. 3: 4. The text states “The preceding experiments showed that concentrations of 10 μM or lower of FUdR were more efficient than AraC in reducing glia cell numbers” this is only true in the case of P3-4 cultures, but not for P0-2 cultures.

Authors: We have updated the respective parts of the text.

Reviewer #1: Fig. 3: 5. The discussion states “Judged by morphology, the further unlabeled cells in our cultures might be GFAP-negative astrocytes [28], activated microglia, less well stained with TMEM119, or oligodendrocytes and their precursors.” This can be determined using other markers for these cell types, including NG2 for OPCs, A2B5 for pre-oligos, and Iba1 or CXCL4 for microglia. Also, TMEM119 is developmentally regulated and doesn’t arise until later postnatal time points up to P14 so some negative cells can exist, whereas Iba1 or CXCL4 will identify all microglia (Bennett, ML et al., 2016, PNAS).

Authors: We have removed the TMEM119 experiments and removed all speculations about the cell type of the unlabeled cells. Unfortunately, we do not have the capacity to repeat these experiments with the suggested markers at the moment. However, we really appreciate the advice and might investigate the effects of cytostatics on microglia in more detail in future.

Reviewer #1: Fig. 4: 1. The text states “There was, however, an insignificant tendency for the highest absorbance per cell in the cultures treated with 50 μM FUdR, which contain the highest number of neurons (see Fig 2D).” this isn’t supported by the data in 2B which quantifies numbers of neurons.

and

Reviewer #1: Fig. 4: 2. The authors report that 50 μM FUdR treatment condition causes increased formazan production which does not reach statistical significance. They claim that this may be due to increased mitochondrial activity in neurons due to Na+ load from higher neuronal cell numbers. As stated previously, the data does not support that there is an increase in neuronal cells, however there is indeed a reduction in glia. Including an additional analysis of the higher FUdR concentrations will be helpful to see if any changes to overall cell viability are occurring. Furthermore, investigating the effects on astrocytes as a whole is essential, including the possibility of excitotoxicity due to reduced astrocyte coverage.

Authors: We re-did the MTT and patch clamp experiments with 75 µM FUdR and AraC on p0-2 as it was suggested by the reviewers to analyze higher FUdR concentrations in the MTT assays and to present patch clamp data from treatment with both cytostatics. The figures now only show the results of these experiments and the body text was updated respectively.

Reviewer #1: Fig. 5: The authors only investigated the effects of FUdR on the Na+ currents in these cultures but did not look at the effect of AraC. The text is lacking in rationale for why they left this out. It is especially relevant given that the field believes AraC to be cytotoxic at higher doses, where this paper proves this not to be the case.

Authors: We performed a new series of patch clamp recordings. For reasons of consistency, we exchanged 100 µM FUdR for 75 µM FUdR and added 75 µM AraC as treatment condition. The text has been updated to match the new data.

Reviewer #1: Fig. 1: The text says that you look at 6 μM for AraC and 25 μM for FUdR but then figure 1 shows all concentrations and throughout the rest of the text, 6 μM for AraC is never mentioned again.

Authors: We removed the data for 6 µM AraC and 25 µM FUdR from the manuscript as during the time course of the experiments we only investigated these concentrations with one of the cytostatics but not both. Since the removal of these two treatment conditions did not change the overall findings of the experiments, we think that the readability of the manuscript benefits from a reduction of the data at this point.

Reviewer #1: Fig 2: The text says that AraC is in orange and FUdR is in purple but in 2A, the text shows AraC in purple and FUdR in orange.

Authors: This has been corrected.

Reviewer #1: Fig 3: Fig 3E, pink arrow on bottom right is pointing to a GFAP positive cell.

Authors: This figure has been removed as it was part of the TMEM119 experiments, which were removed from the manuscript as explained above.

Reviewer #1: Fig 4: Figure legend for 4A says “shown as circles” but there are no circles.

Authors: This has been corrected.

 

Reviewer #2: In financial disclosure: Is stated that the author(s) received no specific funding for this work, however in acknowledge section some funding is mentioned

Authors: We have removed the funding information from the Acknowledgment and asked the editor in the cover letter to add this information to the financial disclosure section. It is not possible for us to change the original submission form now (Note from the editor: ‘Please include your amended statements within your cover letter; we will change the online submission form on your behalf’).

Reviewer #2: The cells were manually counted, and the cell numbers is key to the conclusions of the paper, the reviewer would suggest either an automatic cell counting (several plugins published for Image J allow automatic cell counting), a clear description of how the analyzer was blind to the culture condition, or ideally both

Authors: Data analysis has been repeated. This time CellProfiler and CellProfiler Analyst have been used to count the cells. Unlike suggested, Image J has not been used for automated data analysis, as not only automated counting but also classification was needed. Automated classification on our images required machine learning algorithms. With deepImageJ an ImageJ plugin for this has been published at the end of last year (Sept 30th 2021). However, as it was completely new and we had no prior experiences using the plugin, but have worked with CellProfiler and its extension CellProfilerAnalyst before, we opted for these open-source programs to automate our data analysis. A detailed description of the automated analysis as well as the used CellProfiler pipelines and result tables can be found in the supplementary material. 

Reviewer #2: Add was found at the end of first paragraph of results… however, no significant differences of total cell count between any of the treated cultures was found (p>0.05).

Authors: This has been corrected.

Reviewer #2: Figure legend 1. Should indicate how many independent cultures were analyzed instead of numbers of dishes

Authors: This has been corrected.

Reviewer #2: Pooling data together from different drug concentrations is not a valid comparison and should be removed

Authors: These parts in the text have been removed.

Reviewer #2: I recommend changing the color of the ICC-IF since red, green colour blindness is the most common form of colour vision deficiency

Authors: We have changed red to magenta in the ICC images to make our figures more accessible for readers with color vision impairments.

Reviewer #2: Page 15, line 274 Treatment with 4 uM FUdR, however, inhibited astrocyte proliferation more efficiently than AraC in cultures prepared from P3-4 rats, resulting in a reduction of astrocytes to about 20% of the control level.

It needs to specify P3-4 since the difference is not significant for P0-2 culture, also eliminate already at a very low concentration of the cytostatic as the concentration is already mentioned.

Authors: This has been updated.

Reviewer #2: Page 15, lines 284-286. There was “almost” no difference between the two age groups… or there was a significant age-dependent effect??

Authors: This sentence has been rewritten to improve readability.

Reviewer #2: Page 16, line 287. In P0-2 cultures with increasing concentration of antimitotic agent the neuron to glia… be specific, replace antimitotic agent by FUdR

Authors: It is now specified that this was the case in FUdR-treated cultures.

Reviewer #2: Using the same color for two different markers (in this case, TMEM119 and beta3-tubulin) is not acceptable. The authors need to repeat this staining if the same secondary was used or image the ICC-IFs again using a microscope with more cubes or lasers to be able to separate the 4 markers used.

Authors: We agree. The experiments were conducted to show that almost all of the previously unlabeled cells were likely microglia. If nearly all GFAP- and beta-III-tubulin-negative (unlabeled) cells had disappeared after an additional TMEM119 staining, it would have indicated that most of the unlabeled cells in Fig2A must have been microglia. We were aware that this is questionable for quantification and therefore did not quantify cells stained with TMEM119 and beta-III-tubulin in these images. However, we completely understand that this experiment is flawed and confusing and have removed it from the manuscript.

Reviewer #2: Re-write page 17 lines 318-320 to improve readability. The sentence “To assess cell…” is repetitive and unclear

Authors: This sentence has been rewritten to improve readability.

Reviewer #2: Remove sentence “There was, however, an insignificant tendency…” in page 17 lines 328. Insignificant tendency = no difference

Authors: We agree with the reviewer and have removed this sentence.

Reviewer #2: Figure legends 4 and 5 states “shown as circles/single data point plotted as circles” no circles are shown in the figures

Authors: This has been corrected.

Reviewer #2: Page 23 lines 452-455, and 456-457. The authors make conclusions based on no significative effects. Slightly higher (p>0.1856), slightly improved, tendentially preserved… These need to be removed from the text as are not statistical significative effects and only distract the reader of the main conclusion of the paper that is clearly stated in Page 21 lines 416-418.

Authors: We agree with the reviewer and have removed these passages.

Reviewer #2: In conclusion, page 25, line 473 replace “minimize interaction with glia” for “minimize glia content” and “highly purified neuronal cultures” for “highly enriched neuronal cultures”

Authors: We changed the text as proposed by the reviewer.

Reviewer #2: Page 25, line 489 replace “experimental conditions require more astrocytes to be present in the neuronal vicinity, low…” for “experimental conditions require mix cultures, low…”

Authors: We changed the text as proposed by the reviewer.

---

## [Decision Letter · Decision Letter 1]

23 Feb 2022

Adjusting the neuron to astrocyte ratio with cytostatics in hippocampal cell cultures from postnatal rats: A comparison of cytarabino furanoside (AraC) and 5-fluoro-2’-deoxyuridine (FUdR)

PONE-D-21-30899R1

Dear Dr. Leßlich,

We’re pleased to inform you that your manuscript has been judged scientifically suitable for publication and will be formally accepted for publication once it meets all outstanding technical requirements.

Please make sure that during the final preparation of your manuscript, you incorporated all textual revisions requested by Reviewer #2.

Kind regards,

Alexander A. Mongin, Ph.D.

Academic Editor

PLOS ONE

Additional Editor Comments (optional):

Reviewers' comments:

Reviewer's Responses to Questions

**Comments to the Author**

1. If the authors have adequately addressed your comments raised in a previous round of review and you feel that this manuscript is now acceptable for publication, you may indicate that here to bypass the “Comments to the Author” section, enter your conflict of interest statement in the “Confidential to Editor” section, and submit your "Accept" recommendation.

Reviewer #1: All comments have been addressed

Reviewer #2: (No Response)

2. Is the manuscript technically sound, and do the data support the conclusions?

Reviewer #1: Yes

Reviewer #2: Yes

3. Has the statistical analysis been performed appropriately and rigorously? 

Reviewer #1: Yes

Reviewer #2: Yes

4. Have the authors made all data underlying the findings in their manuscript fully available?

Reviewer #1: Yes

Reviewer #2: Yes

5. Is the manuscript presented in an intelligible fashion and written in standard English?

Reviewer #1: Yes

Reviewer #2: Yes

6. Review Comments to the Author

Reviewer #1: (No Response)

Reviewer #2: The authors did a good job addressing my comments on the first round of revision. Now I only suggest minor changes:

- Last two sentences of methods immunocytochemistry are repetitive, erase one.

- For consistency use the same format for the p values through the paper. Currently sometimes p<0.001 is used, others p=exact value, others p<0.006, p<0.004 ….

- Page 17, line 330 … however, the effect was not significant (p=0.262), page 17, line 334-336 Moreover, there was a significant difference of the … (p=0.24). Why p=0.262 is not significant and p=0.24 is it?

- Page 25, line 487, erase including microglia since TMEM119 staining was removed

7. PLOS authors have the option to publish the peer review history of their article (what does this mean?). If published, this will include your full peer review and any attached files.

Reviewer #1: No

Reviewer #2: No

---

## [Editor Report · Acceptance letter]

28 Feb 2022

PONE-D-21-30899R1 

Adjusting the neuron to astrocyte ratio with cytostatics in hippocampal cell cultures from postnatal rats: A comparison of cytarabino furanoside (AraC) and 5-fluoro-2’-deoxyuridine (FUdR) 

Dear Dr. Lesslich:

I'm pleased to inform you that your manuscript has been deemed suitable for publication in PLOS ONE. Congratulations! Your manuscript is now with our production department. 

Kind regards, 

on behalf of

Dr. Alexander A. Mongin 

Academic Editor

PLOS ONE